mathematical physics/mathematical modelling

prisoner's dilemma game, hawk-dove game, stag hunt game, weight-lifting game, changing social structure

**Authors for correspondence:**
Jin Yoshimura
e-mail: yoshimura.jin@shizuoka.ac.jp
Takuya Okabe
e-mail: okabe.takuya@shizuoka.ac.jp

†Equally contributed.

# Improving environment drives dynamical change in social game structure

Erika Chiba[1,†], Diane Carmeliza N. Cuaresma[2,3,†], Jomar F. Rabajante[3,4], Jerrold M. Tubay[3], Maica Krizna Areja Gavina[3], Tatsuki Yamamoto[5], Jin Yoshimura[1,6,7,8,9,10], Satoru Morita[1,2], Hiromu Ito[6,11] and Takuya Okabe[1,12]

[1]Faculty of Engineering, Shizuoka University, Hamamatsu, Shizuoka 432-8561, Japan
[2]Graduate School of Science and Technology, Shizuoka University, Hamamatsu, Shizuoka 432-8561, Japan
[3]Mathematics Division, Institute of Mathematical Sciences and Physics, University of the Philippines Los Baños, College, Laguna 4031, Philippines
[4]Faculty of Education, University of the Philippines Open University, College, Laguna 4031, Philippines
[5]Star Micronics Co., Ltd., Shizuoka, Shizuoka 422-8654, Japan
[6]Department of International Health and Medical Anthropology, Institute of Tropical Medicine, Nagasaki University, Nagasaki 852-8523, Japan
[7]Department of Biological Sciences, Tokyo Metropolitan University, Hachioji, Tokyo 192-0397, Japan
[8]The University Museum, University of Tokyo, Bunkyo-ku, Tokyo 113-0033, Japan
[9]Marine Biosystems Research Center, Chiba University, Uchiura, Kamogawa, Chiba 299-5502, Japan
[10]Department of Environmental and Forest Biology, State University of New York College of Environmental Science and Forestry, Syracuse, NY 13210, USA
[11]Department of Environmental Sciences, Zoology, University of Basel, Basel 4051, Switzerland
[12]Graduate School of Integrated Science and Technology, Shizuoka University, Hamamatsu, Shizuoka 432-8561, Japan

JFR, 0000-0002-0655-0893; MKAG, 0000-0002-7654-005X; JY, 0000-0003-1610-1386; SM, 0000-0001-5219-6218; HI, 0000-0001-9350-0546; TO, 0000-0001-7518-5837

The development of cooperation in human societies is a major unsolved problem in biological and social sciences. Extensive studies in game theory have shown that cooperative behaviour can evolve only under very limited conditions or with additional complexities, such as spatial structure. Non-trivial two-person games are categorized into three types of games, namely, the prisoner's dilemma game, the chicken game and the stag hunt game. Recently, the weight-lifting game has been shown to cover all five games depending on the success probability of weight lifting, which include the above three

games and two trivial cases (all cooperation and all defection; conventionally not distinguished as separate classes). Here, we introduce the concept of the environmental value of a society. Cultural development and deterioration are represented by changes in this probability. We discuss cultural evolution in human societies and the biological communities of living systems.

## 1. Introduction

Cooperative behaviour has been found extensively in both human societies and social animals [1–4]. Studies in game theory have offered insights into the origin of such cooperation, e.g. in terms of public goods games [5–14]. However, the evolution of society after the establishment of cooperation [15] has rarely been explored. Game theory has been developed to investigate competitive interactions among players (individuals or companies) [1,16,17]. The most commonly studied are the prisoner's dilemma (PD) game and the hawk-dove game, the latter generally known as the chicken game (CH) [5,12,18]. Mathematical analysis of pairwise games shows that selfish players usually dominate altruistic (cooperative) players. Thus, it is difficult to evolve a cooperative society. To explain the cooperative behaviour of animals, Maynard Smith [3] introduced the game theory to biology and developed the concept of evolutionarily stable strategies. Past studies aimed to explain the evolution and maintenance of cooperation. Various characteristics have been proposed to explain the advantage of cooperation, which are broadly categorized into five rules: (i) kin selection, (ii) direct reciprocity, (iii) indirect reciprocity, (iv) network reciprocity, and (v) group selection [19,20].

While games have been considered separately so far (but see [21]), the weight-lifting game was proposed recently [22,23]. The weight-lifting game is a grand unified game covering all kinds of five pairwise games, i.e. the PD, the CH, the stag hunt game (SH), trivial cooperation (TC) and trivial defection (TD) [22]. It should be remarked that the two-by-two game is conventionally considered as consisting of four classes because the last two trivial cases (also called C-dominant and D-dominant Trivials [24]) are not distinguished for their common characteristic of no dilemma [24–28]. Here, we follow the latter terminology, namely C-dominant Trivial (CT) and D-dominant Trivial (DT) in place of TC and TD, respectively [24,28]. In the weight-lifting game, the probability of success or each player's payoff depends on the number of cooperators in the game. The uniqueness of this game is that we can cover all the pairwise games in a single-game framework by changing the success probabilities, which are considered as input parameters. In the present study, we apply the weight-lifting game model to discuss the evolution of a cooperative society by representing the success probability as a function of a state variable, called environmental value. With this variable, we intend to express the environmental conditions experienced by society.

Many empirical studies have been carried out in the framework of conventional game theory [29,30]. To account for empirical results from controlled studies, various modifications and generalizations of game theory and expected utility theory have been put forward [29–36]. For example, behavioural game theory takes account of emotions and limited foresight of average people [33,34]. Recently, a lab-in-the-field experiment was made to study individuals' behaviour when facing different situations corresponding to various dyadic games [21]. This study supplied interesting empirical data for simulating societies in the context of game theory. They found that the subjects conform to a limited number of behavioural phenotypes (envious, optimist, pessimist and trustful) [21]. The present paper is founded on the conventional framework of game theory, taking a similar aspect to those taken by utility functions in economics [35]. However, unlike the traditional studies that focus on the results of an individual game, we study what kind of game is more likely to be provided depending on the environmental state of a dynamically changing society. This type of social investigation has not been made previously. In relation to the current state of the literature in game theory, the success probability of the weight-lifting game may be compared with context-related parameters in various specific situations, such as relatedness $r$ in the kin selection, the probability of mutual encounter $w$ in direct reciprocity, the probability $q$ of knowing someone's reputation in indirect reciprocity, the average number of neighbours $k$ per individual in network reciprocity and so on [19,27].

There are various kinds of cooperative societies. For example, Eskimos living in the northern circumpolar region experience a very harsh environment throughout the year. People living in a desert also face severe conditions and risks to their lives resulting from the shortage of water. In temperate regions, environmental conditions are so mild that such life-threatening problems are not found. Therefore, many developed human societies have been established in these regions. To express the severity of the environment, we introduce an environmental value as a measure of the wealthiness of the society.

Changes in the environmental value will affect the survival of individuals. Under severe conditions, the survival of individuals is very difficult, even if all individuals help each other. We interpret this situation as follows: (i) in a severe environment, the success probability of the weight-lifting game is low even when both players cooperate. (ii) As the environmental value improves, this probability increases. (iii) Further improvement of the environmental value saturates the probability up to the maximum value (unity) independently of the players' strategies. Thus, we can view change in the environment as a dynamical shift in the environmental value, which is translated into the success probability of the weight-lifting game.

## 2. Model and results

In the weight-lifting game, baggage is carried by two players randomly selected from an unlimited well-mixed population [22,23]. Each player chooses one of two strategies: cooperation (C) in carrying the baggage by paying a cost ($c \geq 0$) or defection (D) without any cost. If the baggage is carried successfully, each player obtains a net benefit ($b > 0$) irrespective of his/her strategy (figure 1$a$). Let $p_{n_c}$ be the probability of success, where $n_c$ ($= 0, 1, 2$) is the number of cooperators ($0 \leq p_{n_c} \leq 1$). We introduce two parameters: $\Delta p_1 = p_1 - p_0$ ($0 \leq \Delta p_1 \leq 1$) and $\Delta p_2 = p_1 - p_0$ ($0 \leq \Delta p_1 \leq 1$). These parameters express increments of the success probability caused by the presence of one cooperator. The difference between the success probabilities of two cooperators and no cooperators is $\Delta p_1 + \Delta p_2$ (figure 1$b$). The payoff matrix of this game is represented in terms of the expected gains for the two strategies (figure 1$c$).

The success probability also depends on the environmental value $E$, i.e. $p_{n_c} = p(E, n_c)$ ($0 \leq E \leq 1$). This function, $p(E, n_c)$, should satisfy the following conditions: it is an increasing function of $n_c$ as well as $E$ and varies between $p(0, n_c) = 0$ and $p(1, n_c) = 1$. In the extreme cases, $E = 0$ and $1$, the success probability takes the lowest and the highest value in the poorest and richest society, respectively, regardless of the number of cooperators $n_c$. For simplicity, we adopt

$$p(E, n_c) = E^{\delta^{n_c - 1}},$$

where $\delta$ is a small positive number ($0 \leq \delta \leq 1$). Note that $p(E, n_c) = E$ for $\delta = 1$ independently of $n_c$. The smaller $\delta$ is, the stronger the $n_c$ dependence of $p(E, n_c)$. In figure 2$a$, the probability $p_{n_c} = p(E, n_c)$ for $\delta = 1/3$ is plotted against $n_c$ for $E = 0, 0.25, 0.5$, and $0.75$. In figure 2$b$, the $p_{n_c}$ values for $n_c = 0, 1$ and $2$ are shown on a line of unit length for $E = 0, 0.25, 0.5, 0.75$ and $1$ ($\delta = 1/3$). It should be remarked that the qualitative behaviour of $p(E, n_c)$, but not its particular form ($E^{\delta^{n_c - 1}}$), is important for the following results (electronic supplementary material). Figure 3 shows how the game structure changes as the environmental value $E$ varies. Figure 3$a,b$ shows the trajectories in the $\Delta p_1$-$\Delta p_2$ plane for $c/b = 1/2$ and $1/3$, respectively. Figure 3$c$ shows the trajectories in the $E$-$c/b$ plane for $c/b = 1/2$ (solid) and $1/3$ (dashed). In figure 3$a$–$c$, the parameter $\delta$ is $\delta = 1/3$. Similarly, the panels in figure 3$d$–$f$ represent $\delta = 1/5$. The results indicate that CT occurs only when the benefit-to-cost ($b/c$) ratio is large ($c/b$ is small) and $E \approx 0.5$ (figure 3$b,e$).

Game structure varies depending on the $b/c$ ratio (figure 3$c,f$). For a fixed value of $b/c$, for instance, a game structure may appear as DT-PD-DT (solid arrows in figure 3$c$), DT-PD-SH-CH-PD-DT (dashed arrows in figure 3$c$), DT-PD-SH-PD-CH-PD-DT (solid arrows in figure 3$f$) and DT-PD-SH-CT-CH-PD-DT (dashed arrows in figure 3$f$). It is generally expected that the $b/c$ ratio will increase in an accelerated manner as the environmental value increases to the maximum ($E = 1$). The game-space trajectory for $b/c = 3/(\beta - E)^2$ is plotted in figure 4$a,b$ for $\beta = 1.5$ and $1.3$ ($\delta = 1/5$). Game structure varies according to the sequence DT-PD-SH-CT-CH-PD-DT, while some of the intermediate games may be skipped or modified depending on specific details of the $b/c$ trajectory (figure 4).

## 3. Discussion

The results of the proposed model are robust because these results originate from the general assumption that the probability of success increases from near 0 to 1 (unity) (figures 2 and 3; electronic supplementary material, figures S1, S3 and S5). Note that the increasing probability is expressed as a left-to-right trajectory in figure 3$c,f$ and electronic supplementary material, figures S2$c$, S4$c$ and S6$c$. This probability assumption simply expresses that a wealthy society has more success than a poor society. The assumption of increasing probability (0 to 1) can be violated in some irregular societal

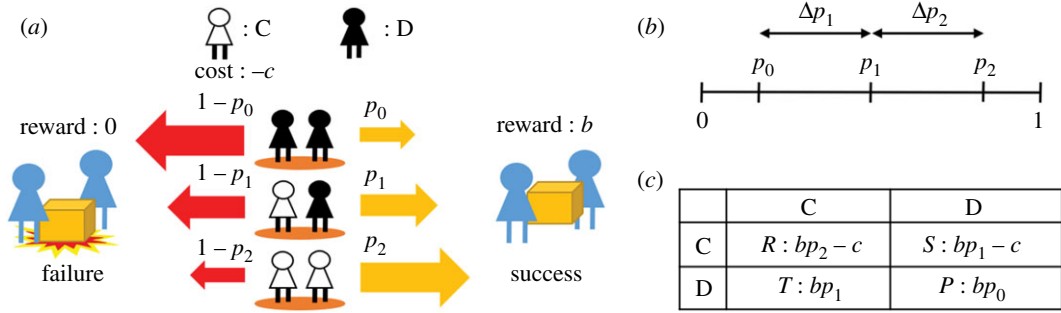

**Figure 1.** The weight-lifting game. (*a*) Two players lift the baggage (weight). A cooperator (C, white) pays a cost $c$, while a defector (D, black) does not. Each player receives either a reward $b$ or nothing depending on whether the lifting is successful. The success probability $p_{n_c}$ depends on the number of cooperators ($n_c = 0$, 1 and 2). (*b*) We define $\Delta p_1$ and $\Delta p_2$ as the differences $p_1 - p_0$ and $p_2 - p_1$, respectively. Each of $\Delta p_1$, $\Delta p_2$ and $\Delta p_1 + \Delta p_2$ takes a numeric value between 0 and 1. (*c*) The payoff matrix of the weight-lifting game.

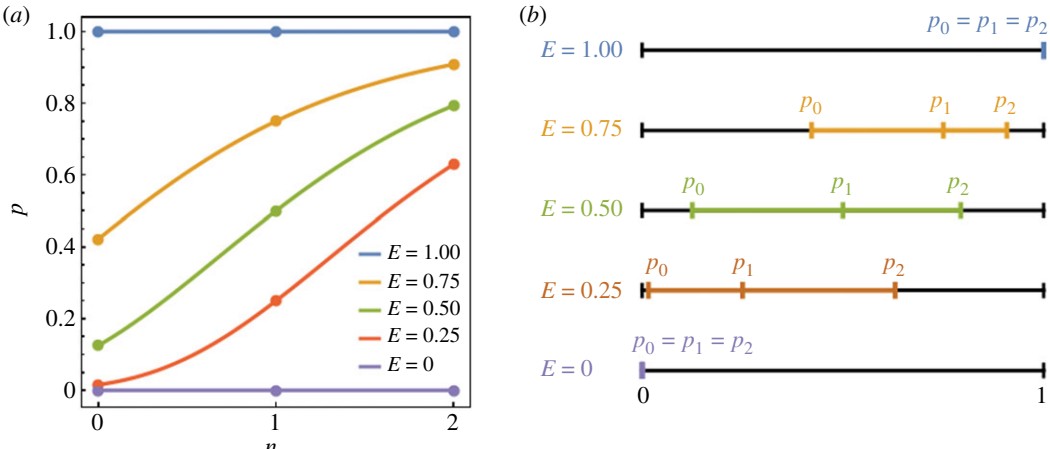

**Figure 2.** The success probability $p(E, n_c)$. (*a*) $p(E, n_c)$ is plotted against $n_c$ for $E = 0$, 0.25, 0.5, 0.75 and 1 ($\delta = 1/3$). (*b*) $p_{n_c} = p(E, n_c)$ for $n_c = 0$, 1 and 2 are shown on a line of unit length for $E = 0$, 0.25, 0.5, 0.75 and 1 ($\delta = 1/3$).

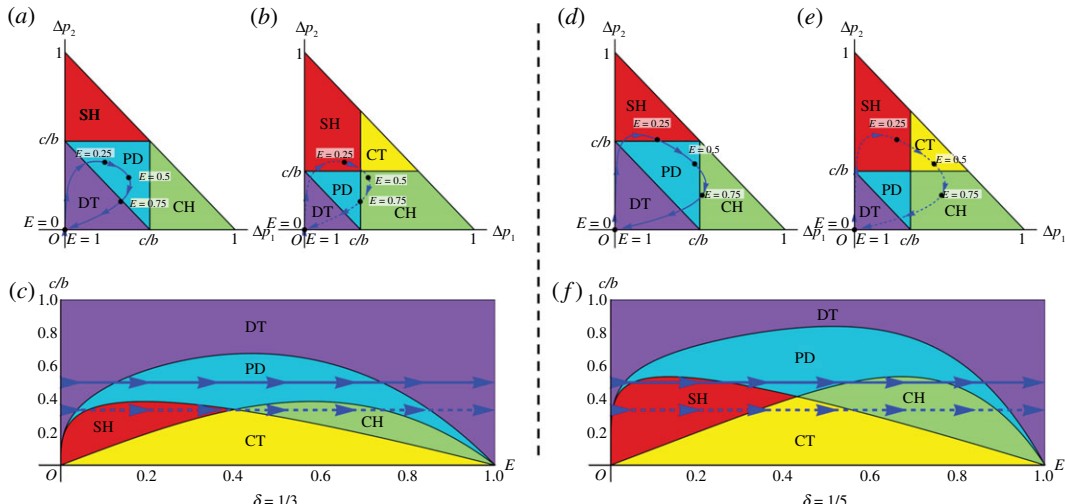

**Figure 3.** Trajectory in the game phase diagram as the environmental value $E$ varies from 0 to 1. (*a*) $c/b = 1/2$ and $\delta = 1/3$. (*b*) $c/b = 1/3$ and $\delta = 1/3$. (*c*) $c/b = 1/2$ (solid) and 1/3 (dashed) for $\delta = 1/3$. (*d*) $c/b = 1/2$ and $\delta = 1/5$. (*e*) $c/b = 1/3$ and $\delta = 1/5$. (*f*) $c/b = 1/2$ (solid) and 1/3 (dashed) for $\delta = 1/5$. The coloured areas represent all kinds of pairwise games, i.e. the prisoner's dilemma (PD: blue), the chicken game (CH: green), the stag hunt game (SH: red), D-dominant trivial (DT: purple) and C-dominant trivial (CT: yellow).

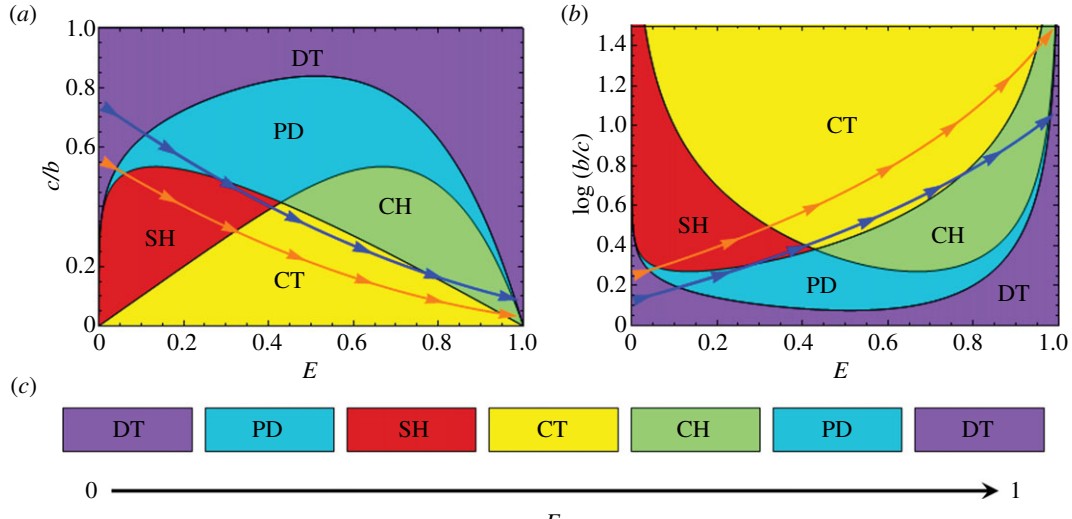

**Figure 4.** Change in game structure as the environmental value $E$ varies. ($a$) Trajectories in the $E$–$c/b$ diagram. ($b$) Trajectories in the $E$–$b/c$ diagram. $b/c = 3/(\beta - E)^2$ for $\beta = 1.5$ (blue) and 1.3 (orange) ($\delta = 1/5$). ($c$) How the game varies as the environmental value $E$ changes from 0 to 1. The coloured areas represent all kinds of pairwise games, i.e. the prisoner's dilemma (PD: blue), the chicken game (CH: green), the stag hunt game (SH: red), D-dominant trivial (DT: purple) and C-dominant trivial (CT: yellow).

conditions. We should also relax this probability condition when we simply compare various present societies simultaneously.

We demonstrated that the structure of the weight-lifting game dynamically changes as the environmental value varies. In a prospering society, all individuals gain benefits irrespective of their behaviour. For example, members of some rich nations receive a salary without paying tax. Thus, the game and rewards depend on the cultural state (richness) of the society. When the environmental value is low, we found that the structure of the game may become that of the stag hunt game, which has rarely been studied [37]. In a real society, this should correspond to a premature state, in which cooperation has not yet been fully established. Moreover, cooperation is not sustained if the $b/c$ ratio is too low. If we consider that the benefit of cooperation underlies the establishment of societies, true sociality starts from CT with a relatively high $b/c$ ratio. The cooperative society should then proceed via the CH and the PD game to DT, at which point the society collapses.

The present result that the SH game precedes CT has implications for the evolution of social structure [37]. In a primitive society with SH game structure, all members being cooperative as well as all members being defective are Nash equilibria. Therefore, some incentive toward cooperation is necessary because a cooperative society does not emerge spontaneously. Since the $b/c$ ratio is not high enough, communities without any cooperation have a good reason to persist in a stable manner. However, once a cooperative society is established for some reason, it will persist, as it is protected against the intrusion of a small number of defectors. Location (interaction with neighbours), signals (transmission of information) and association (the formation of social networks) are suggested factors promoting the establishment of payoff-dominant (cooperative) societies [37]. Another possible reason may be accidental changes in the environment. In any case, the SH state is a necessary but insufficient condition for a cooperative society. As the $b/c$ ratio grows, the condition becomes sufficient, and a true cooperative society in the CT state is established. In future, this dynamical shift in game classes may be translated into the dynamical change in a social dilemma situation (phase plane of gamble-intending dilemma (GID) and risk-averting dilemma (RAD) [27]) [24]. Then, the stability of behavioural phonotypes in humans may be interpreted as the optimal solution under dynamical societal changes. Thus, the rise-and-fall (prosperity and decline) of human societies may be understood as historical changes in the game structure of the grand unified game.

Cooperative societies are not unique to humans and instead broadly relevant to the history of life from its origin to the present time [38]. In the beginning of life, approximately 4 billion years ago, multiple species of chemotrophic bacteria lived cooperatively by forming symbiotic microbial mats (often called bacterial consortia) [39–41]. Each bacterium played an indispensable role in the chemotrophic cycle, i.e. obligate symbiosis. Even at the present time, there exist such chemotrophic cycles without photosynthesis in hot springs, deep-sea vents and soil. In terms of the present model, the origin of life begins with the CT state, which is sustained because independence from cooperation signifies the end of its existence.

On a geological timescale, cyanobacteria evolved hundreds of millions of years later, forming the photosynthetic cycle and winning alone. This is the first time that organisms escaped the non-oxygen-based chemical synthesis cycle. It is not clear whether cyanobacteria evolved alone (from a single bacterium) or as symbionts. It is believed that cyanobacteria have filled the Earth's shallow waters for hundreds of millions of years, thereby transforming the Earth into an oxygenated planet [42]. Photosynthesis is a complex cycle that involves multiple steps, known as the Krebs cycle, and cyanobacteria may have evolved through the symbiosis and integration of many bacteria. Even at the present time, some cyanobacteria coexist with fungi to form lichens in order to survive in poor environments, such as in air, on rocks and in polar regions.

Subsequently, eukaryotes evolved through prokaryotic symbiosis [43]. The vestige of prokaryotes remains as subcellular organelles in eukaryotes. This symbiosis is also an obligate symbiosis, in which betraying others does not benefit oneself. The Ediacaran organisms seen in the late Proterozoic were multicellular eukaryotes and are also considered to have formed obligate symbioses. During the Cambrian Explosion, in the early Palaeozoic Cambrian 544 million years ago, diversification and adaptive radiation occurred, with tissue/organ differentiation in invertebrates such as trilobites [44,45]. These multicellular organisms were giant symbiotic organisms where countless cells (tissues and organs) cooperated by absolute symbiosis, the birth of a multicellular individual. In this way, organisms have evolved since the origin of life as obligate symbiotic systems. Obligate symbiosis from the viewpoint of the geological history of life is thus considered to be maintained in the CT state under the present theory.

While betrayal is not advantageous in absolutely symbiotic systems, it is commonly seen in highly organized social systems similar to human societies. For example, in colonies of the eusocial insect *Pristomyrmex punctatus*, eggs are laid and maintained by worker ants, which are parasitized by queen ants of different strains. The parasitic queens lay a large number of eggs but do not care for them. As a matter of course, the parasitized colony eventually collapses, as it is exploited by non-working queens. Unparasitized colonies divide to form new colonies fast enough to cover the loss of parasitized colonies. Here, the relationship between worker and queen ants may be understood as an arms race or as being maintained for some unknown reason [46]. Similar social parasitism has been described in cuckoos (*Cuculus canorus*) and blackbirds (*Turdus merula*). This type of social 'free-rider' system is also found in a group of hunting wasps (parasitoid wasps), where free-rider species parasitize other species, often closely related to the free-rider [47,48]. In such cases, the parasitic members (parasitic birds and wasps) are betraying, so it would not be possible to apply this game directly. However, this transition may be regarded as a shift from CT symbiosis to CH and DT (figures 3 and 4) because the societies (communities of birds and wasps) could afford the reproductive burden of the parasitic species.

In this paper, we demonstrate the relationships between the environment experienced by a society and plausible pairwise games based on the success probability of weight lifting. This relationship shows that certain types of games occur frequently while others occur less frequently depending on environmental conditions. The rise-and-fall (prosperity and decline) of a society can be viewed as a decay process from CT to DT via CH and PD. The coexistence of organisms since the origin of life is caused by the changes in environmental conditions. Thus, the history of life may be viewed as an interactive evolutionary game under changing conditions. The transition of games played by different kinds of members may be investigated in future.

It is noted that this study is but a first attempt so that the current form does not allow us to test it empirically in a direct manner. The model assumption that the success probabilities change depending on the environmental conditions corresponds to the numerous historical facts of rise-and-fall in human culture, as a conceptual way. The direct verification of the model is, however, very difficult because such historical rise-and-fall is difficult to quantify. A more concrete verification of the current model is a future step. One possible method is to compare the human responses against the weight-lifting game before and after a radical societal change over a short time period, e.g. before and after the COVID-19 pandemic. Another similar approach is to compare various societies under different conditions, e.g. the comparison of the responses of people among various countries.

Data accessibility. This paper uses no data and all the figures can be drawn from the equations provided in it. Correspondence and requests for materials should be addressed to okabe.takuya@shizuoka.ac.jp.

Authors' contributions. E.C., T.Y., J.Y. and T.O. conceived the study and developed the original model. D.C.N.C., J.F.R., J.M.T., M.K.A.G., H.I., S.M. and T.O. analysed and finalized the model. D.C.N.C., H.I., J.Y. and T.O. developed the practical implications. E.C., J.Y. and T.O. wrote the draft manuscript. All revised and finalized the manuscript.

Competing interests. The authors have no conflicts of interest to declare.

Funding. This work was partly supported by the Japan Society for the Promotion of Science (JSPS) KAKENHI (grant nos. 17J06741,17H04731 and 19KK0262 to H.I.; grant no. 18K03453 to S.M.; grant nos. 15H04420 and 26257405 to J.Y.; grant no. 21K12047 to T.O.).

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
