## [Peer Review File · Royal Society Open Science]

Review History

RSOS-201166.R0 (Original submission)

Review form: Reviewer 1

Is the manuscript scientifically sound in its present form?

Yes

Are the interpretations and conclusions justified by the results?

Yes

Is the language acceptable?

Yes

Do you have any ethical concerns with this paper?

No

Have you any concerns about statistical analyses in this paper?

No

Recommendation?

Accept with minor revision (please list in comments)

Comments to the Author(s)

This work establishes a new 2-payer & 2-strategy game. The authors' main claim is that the proposed game has 5 classes, which is more than the conventional 2 by 2 game having 4 game-classes; PD, Chicken, SH and Trivial (non-dilemma game). Their game has PD, Chicken, SH, D-dominant Trivial (that is called Trivial Defection by the authors; which is not appropriate naming; which should be call, as I said, D-dominant Trivial) and C-dominant Trivial (that is called Trivial Cooperation). Although the authors may think it is a significant finding, it seems, at least to me, quite trivial, say, 'literally' trivial honestly. The reason is as below. Referring to the game payoff structure; Fig. 1, depending on the presumed probabilistic parameter; p_0 and p_2 , the order of the diagonal elements; $R (=b*p_2 - c)$ and $P (=b*p_0)$ may change, i.e., both cases; $R > P$ and $R < P$ might be possible. That is why, instead of the conventional 2 by 2 game having only (C-dominant) Trivial, their game has two Trivial sub-classes; splitting it into twofold; C-dominant Trivial (if meeting with $R > P$) and D-dominant Trivial (if meeting with $R < P$). It seems extremely natural, and no surprising at all.

Theoretically speaking, with the premise of $R > P$, there must be only 4 game classes in symmetric 2 by 2 game, which is clearly proved from mathematical viewpoint. It is because the degree of freedom is 4 in the game. Yet, if one relaxes the constraint of ' $R > P$ ' by allowing the inverse order; ' $R < P$ ', this relaxation unequivocally adds another degree of freedom. Thus, there are five game classes as the authors insisting. From the principle of physics as well, there is nothing to further explain for what they reported here.

In locally summing so far, I am giving, in a sense, a harsh word to the present contribution. But I wouldn't go so far as saying that I totally deny this work.

Although there is none of substantially big novelty, I can evaluate their work as practically informative. It is because their game extends the conventional 2 by 2 game sphere to broader worlds by introducing another set of probabilistic parameters into the payoff matrix; that are p_0 , p_1 and p_3 . This concept struck me as interesting. What they call 'weight-lifting game' sound interesting and catchy to the audience who is interested in the evolutionary game theory. Also their mathematical handling to define those parameters by $p(E, n_c) = E^{(\Delta^{(n_c - 1)})}$ looks nice, and quite intrigued, which ensures monotonically increasing function of n_c of which slop increases with increase of E except for both ends; $E = 0$ and $E = 1$ that are non-sensitive to n_c . The interpretation of E they posed can be 'environmental' effect, which is also conceivable.

To this end, I see this MS can be welcomed to publication only when the revised MS would respond the following suggestions.

#1.

The authors should further deepen their discussion by referring to what I suggested above; 1st and 2nd paragraphs. On the way, they should cite following literature that relating to the universal concept of dilemma strength, why Trivial being called D-dominant Trivial and other theoretically fundamentals;

Evolutionary Games with Sociophysics: Analysis of Traffic Flow and Epidemics, Springer, 2019.1; ISBN: 978-981-13-2769-8

Fundamentals of Evolutionary Game Theory and its Applications, Springer, 2015.10; ISBN: 978-4-431-54961-1

Tanimoto & Sagara; Relationship between dilemma occurrence and the existence of a weakly dominant strategy in a two-player symmetric game, *BioSystems* 90(1), 105-114, 2007.

Wang et al.; Universal scaling for the dilemma strength in evolutionary games, *Physics of Life Reviews* 14, 1-30, 2015.

Ito et al.; Scaling the phase- planes of social dilemma strengths shows game-class changes in the five rules governing the evolution of cooperation, *Royal Society Open Science*, 181085, 2018.

#2.

As I suggested, their terminology of TD and TC should be replaced with D-dominant Trivial and C-dominant Trivial, of which underpinning was given the two book above from theoretical standpoint.

More precisely, a question of whether it coming to 'Trivial' or non 'Trivial', let' say, it coming to 'Dilemma' situation is whether or not "Nash Equilibrium is consistent with equal (meaning 'egalitarian') social optimal situation". I suggest the authors should mentioned this point somewhere in Discussion part as well.

Review form: Reviewer 2 (Josep Perelló)

Is the manuscript scientifically sound in its present form?

No

Are the interpretations and conclusions justified by the results?

No

Is the language acceptable?

Yes

Do you have any ethical concerns with this paper?

No

Have you any concerns about statistical analyses in this paper?

No

Recommendation?

Major revision is needed (please make suggestions in comments)

Comments to the Author(s)

The authors introduce the concept of environmental value which is potentially interesting. However, the authors should be making a stronger effort to make the approach valuable enough. I list some of the reasons below.

The function to capture the related environmental phenomena is not justified enough. The functional is quite specific in its form, taking a similar aspect to those taken by utility functions in economics. The form is not well justified and the authors relies its choice for the sake of simplicity. The authors does not provide a deep exploration about alternative forms and how other forms may affect the current results. The exploration may offer a stronger sense of generality in the discussion section. The lack of a more careful exploration lowers the impact of the contribution and limits the soundness of the formalism provided.

The authors introduce the idea of the importance of the context and as I said it is a very interesting perspective but it is just qualitatively addressed in the paper. I wonder if the authors can better (and quantitatively, eventually) connect this with current state of the literature in game theory (some are considering specific groups of individuals while others are considering mobility or space as an aspect). This may indeed mean to better connect the results with empirical contexts and this may allow to offer realistic parameters choice. A deeper literature research for this purpose is encouraged.

At some point the authors mentions that "the weight-lifting game enables us to investigate them in a unified manner" and that mentioned "games have been considered separately so far". I agree with the first point and concerning the second point I would encourage the authors to look at and related references: Poncela-Casasnovas, J., Gutiérrez-Roig, M., Gracia-Lázaro, C., Vicens, J., Gómez-Gardeñes, J., Perelló, J., ... & Sánchez, A. (2016). Humans display a reduced set of consistent behavioral phenotypes in dyadic games. *Science advances*, 2(8), e1600451. If I am not mistaken, they directly treat the same games' space and there is a description of existing strategies and there is a data set that can be of interest to the authors.

The abstract ends up with a quite strong sentence: "The current results may offer a solution to unanswered question about the origin of cooperation". The feeling after reading the paper is that the authors are solely able to build a function with two parameters to continuously travel through the parameters phase diagram where games are all connected. The authors should make stronger efforts in the paper to make the whole results convincing.

Decision letter (RSOS-201166.R0)

Dear Professor Yoshimura

The Editors assigned to your paper RSOS-201166 "Improving environment drives dynamical change in social game structure" have now received comments from reviewers and would like you to revise the paper in accordance with the reviewer comments and any comments from the Editors. Please note this decision does not guarantee eventual acceptance.

Please submit your revised manuscript and required files (see below) no later than 21 days from today's (ie 18-Jan-2021) date. Note: the ScholarOne system will 'lock' if submission of the revision is attempted 21 or more days after the deadline. If you do not think you will be able to meet this deadline please contact the editorial office immediately.

on behalf of Prof Miles Padgett (Subject Editor)
openscience@royalsociety.org

Associate Editor Comments to Author:

Thank you for the patience while we sought reviewers - regrettably, it has been unusually hard to find commentators for the work (no doubt in large part owing to COVID). While the reviewers have offered a range of suggestions to ensure your paper reaches a publishable standard, there is a concern that this work does not - in its current form - a particularly distinctive contribution to the literature (relative to earlier works by these authors). RSOS does not require ground-breaking novelty, and the paper appears to be largely technically sound, but the value added to the literature would be increased if the authors can emphasise how this work builds on and is distinct from their earlier work. We'll look forward to receiving your revision in due course.

Reviewer comments to Author:

Reviewer: 1

Comments to the Author(s)

This work establishes a new 2-payer & 2-strategy game. The authors' main claim is that the proposed game has 5 classes, which is more than the conventional 2 by 2 game having 4 game-classes; PD, Chicken, SH and Trivial (non-dilemma game). Their game has PD, Chicken, SH, D-dominant Trivial (that is called Trivial Defection by the authors; which is not appropriate naming: which should be called, as I said, D-dominant Trivial) and C-dominant Trivial (that is called Trivial Cooperation). Although the authors may think it is a significant finding, it seems, at least to me, quite trivial, say, 'literally' trivial honestly. The reason is as below. Referring to the game payoff structure; Fig. 1, depending on the presumed probabilistic parameter; p_0 and p_2 , the order of the diagonal elements; $R (=b*p_2 - c)$ and $P (=b*p_0)$ may change, i.e., both cases; $R > P$ and $R < P$ might be possible. That is why, instead of the conventional 2 by 2 game having only (C-dominant) Trivial, their game has two Trivial sub-classes; splitting it into twofold; C-dominant Trivial (if meeting with $R > P$) and D-dominant Trivial (if meeting with $R < P$). It seems extremely natural, and no surprising at all.

Theoretically speaking, with the premise of $R > P$, there must be only 4 game classes in symmetric 2 by 2 game, which is clearly proved from mathematical viewpoint. It is because the degree of freedom is 4 in the game. Yet, if one relaxes the constraint of ' $R > P$ ' by allowing the inverse order; ' $R < P$ ', this relaxation unequivocally adds another degree of freedom. Thus, there are five game classes as the authors insisting. From the principle of physics as well, there is nothing to further explain for what they reported here.

In locally summing so far, I am giving, in a sense, a harsh word to the present contribution. But I wouldn't go so far as saying that I totally deny this work.

Although there is none of substantially big novelty, I can evaluate their work as practically informative. It is because their game extends the conventional 2 by 2 game sphere to broader worlds by introducing another set of probabilistic parameters into the payoff matrix; that are p_0 , p_1 and p_3 . This concept struck me as interesting. What they call 'weight-lifting game' sound interesting and catchy to the audience who is interested in the evolutionary game theory. Also their mathematical handling to define those parameters by $p(E, n_c) = E^{\Delta(n_c - 1)}$ looks nice, and quite intrigued, which ensures monotonically increasing function of n_c of which slope increases with increase of E except for both ends; $E = 0$ and $E = 1$ that are non-sensitive to n_c . The interpretation of E they posed can be 'environmental' effect, which is also conceivable.

To this end, I see this MS can be welcomed to publication only when the revised MS would respond the following suggestions.

#1.

The authors should further deepen their discussion by referring to what I suggested above; 1st and 2nd paragraphs. On the way, they should cite following literature that relating to the universal concept of dilemma strength, why Trivial being called D-dominant Trivial and other theoretically fundamentals;

Evolutionary Games with Sociophysics: Analysis of Traffic Flow and Epidemics, Springer, 2019.1; ISBN: 978-981-13-2769-8

Fundamentals of Evolutionary Game Theory and its Applications, Springer, 2015.10; ISBN: 978-4-431-54961-1

Tanimoto & Sagara; Relationship between dilemma occurrence and the existence of a weakly dominant strategy in a two-player symmetric game, *BioSystems* 90(1), 105-114, 2007.

Wang et al.; Universal scaling for the dilemma strength in evolutionary games, *Physics of Life Reviews* 14, 1-30, 2015.

Ito et al.; Scaling the phase- planes of social dilemma strengths shows game-class changes in the five rules governing the evolution of cooperation, *Royal Society Open Science*, 181085, 2018.

#2.

As I suggested, their terminology of TD and TC should be replaced with D-dominant Trivial and C-dominant Trivial, of which underpinning was given the two book above from theoretical standpoint.

More precisely, a question of whether it coming to 'Trivial' or non 'Trivial', let' say, it coming to 'Dilemma' situation is whether or not "Nash Equilibrium is consistent with equal (meaning 'egalitarian') social optimal situation". I suggest the authors should mentioned this point somewhere in Discussion part as well.

Reviewer: 2

Comments to the Author(s)

The authors introduce the concept of environmental value which is potentially interesting. However, the authors should be making a stronger effort to make the approach valuable enough. I list some of the reasons below.

The function to capture the related environmental phenomena is not justified enough. The functional is quite specific in its form, taking a similar aspect to those taken by utility functions in economics. The form is not well justified and the authors relies its choice for the sake of

simplicity. The authors does not provide a deep exploration about alternative forms and how other forms may affect the current results. The exploration may offer a stronger sense of generality in the discussion section. The lack of a more careful exploration lowers the impact of the contribution and limits the soundness of the formalism provided.

The authors introduce the idea of the importance of the context and as I said it is a very interesting perspective but it is just qualitatively addressed in the paper. I wonder if the authors can better (and quantitatively, eventually) connect this with current state of the literature in game theory (some are considering specific groups of individuals while others are considering mobility or space as an aspect). This may indeed mean to better connect the results with empirical contexts and this may allow to offer realistic parameters choice. A deeper literature research for this purpose is encouraged.

At some point the authors mentions that "the weight-lifting game enables us to investigate them in a unified manner" and that mentioned "games have been considered separately so far". I agree with the first point and concerning the second point I would encourage the authors to look at and related references: Poncela-Casasnovas, J., Gutiérrez-Roig, M., Gracia-Lázaro, C., Vicens, J., Gómez-Gardeñes, J., Perelló, J., ... & Sánchez, A. (2016). Humans display a reduced set of consistent behavioral phenotypes in dyadic games. *Science advances*, 2(8), e1600451. If I am not mistaken, they directly treat the same games' space and there is a description of existing strategies and there is a data set that can be of interest to the authors.

The abstract ends up with a quite strong sentence: "The current results may offer a solution to unanswered question about the origin of cooperation". The feeling after reading the paper is that the authors are solely able to build a function with two parameters to continuously travel through the parameters phase diagram where games are all connected. The authors should make stronger efforts in the paper to make the whole results convincing.

===PREPARING YOUR MANUSCRIPT===

If you have been asked to revise the written English in your submission as a condition of publication, you must do so, and you are expected to provide evidence that you have received language editing support. The journal would prefer that you use a professional language editing

service and provide a certificate of editing, but a signed letter from a colleague who is a native speaker of English is acceptable. Note the journal has arranged a number of discounts for authors using professional language editing services (<https://royalsociety.org/journals/authors/benefits/language-editing/>).

===PREPARING YOUR REVISION IN SCHOLARONE===

<https://royalsociety.org/journals/authors/author-guidelines/#supplementary-material> to include a suitable title and informative caption. An example of appropriate titling and captioning

may be found at https://figshare.com/articles/Table_S2_from_Is_there_a_trade-off_between_peak_performance_and_performance_breadth_across_temperatures_for_aerobic_sc_ope_in_teleost_fishes_/3843624.

Author's Response to Decision Letter for (RSOS-201166.R0)

See Appendix A.

RSOS-201166.R1 (Revision)

Review form: Reviewer 1

Is the manuscript scientifically sound in its present form?

Yes

Are the interpretations and conclusions justified by the results?

Yes

Is the language acceptable?

Yes

Do you have any ethical concerns with this paper?

No

Have you any concerns about statistical analyses in this paper?

No

Recommendation?

Accept as is

Comments to the Author(s)

The revised MS seems to be acceptable for publication.

Review form: Reviewer 2 (Josep Perelló)

Is the manuscript scientifically sound in its present form?

Yes

Are the interpretations and conclusions justified by the results?

No

Is the language acceptable?

Yes

Do you have any ethical concerns with this paper?

No

Have you any concerns about statistical analyses in this paper?

No

Recommendation?

Reject

Comments to the Author(s)

The authors have answered to most of my concerns and I thank them for this effort.

1) I however still find the paper very disconnected to empirical features or specific situations. The authors have not shown any means to anticipate how the model could become valuable to better interpret real-world situations. There is still no clue on how the model can take real world observation to calibrate somehow the model.

2) Concerning the robustness of the model, the authors should be better explaining why the model is robust enough and why the alternative model supports this statement. The authors have included an obscure sentence in the main paper in relation to these issues and this fact needs further clarification.

Decision letter (RSOS-201166.R1)

Dear Professor Yoshimura

On behalf of the Editors, we are pleased to inform you that your Manuscript RSOS-201166.R1 "Improving environment drives dynamical change in social game structure" has been accepted for publication in Royal Society Open Science subject to minor revision in accordance with the referees' reports. Please find the referees' comments along with any feedback from the Editors below my signature.

Please submit your revised manuscript and required files (see below) no later than 7 days from today's (ie 11-Mar-2021) date. Note: the ScholarOne system will 'lock' if submission of the revision is attempted 7 or more days after the deadline. If you do not think you will be able to meet this deadline please contact the editorial office immediately.

on behalf of Professor Miles Padgett (Subject Editor)
openscience@royalsociety.org

Associate Editor Comments to Author:

The Editors would like you to address the concerns raised regarding both the robustness of the model and also its real-world applications - these are reasonable concerns from the reviewer, and we would like you to include a paragraph or two to provide more exposition here. If you provide this in a revised paper, as well as your response to referees document, the paper may be accepted - but this will be subject to you satisfactorily engaging with these concerns.

Reviewer comments to Author:

Reviewer: 1
Comments to the Author(s)

The revised MS seems to be acceptable for publication.

Reviewer: 2
Comments to the Author(s)

The authors have answered to most of my concerns and I thank them for this effort.

1) I however still find the paper very disconnected to empirical features or specific situations. The authors have not shown any means to anticipate how the model could become valuable to better interpret real-world situations. There is still no clue on how the model can take real world observation to calibrate somehow the model.
2) Concerning the robustness of the model, the authors should be better explaining why the model is robust enough and why the alternative model supports this statement. The authors have included an obscure sentence in the main paper in relation to these issues and this fact needs further clarification.

===PREPARING YOUR MANUSCRIPT===

===PREPARING YOUR REVISION IN SCHOLARONE===

- Any electronic supplementary material (ESM).
- If you are requesting a discretionary waiver for the article processing charge, the waiver form must be included at this step.
- If you are providing image files for potential cover images, please upload these at this step, and inform the editorial office you have done so. You must hold the copyright to any image provided.
- A copy of your point-by-point response to referees and Editors. This will expedite the preparation of your proof.

- Ensure that your data access statement meets the requirements at <https://royalsociety.org/journals/authors/author-guidelines/#data>. You should ensure that you cite the dataset in your reference list. If you have deposited data etc in the Dryad repository, please only include the 'For publication' link at this stage. You should remove the 'For review' link.
- If you are requesting an article processing charge waiver, you must select the relevant waiver option (if requesting a discretionary waiver, the form should have been uploaded at Step 3 'File upload' above).
- If you have uploaded ESM files, please ensure you follow the guidance at <https://royalsociety.org/journals/authors/author-guidelines/#supplementary-material> to include a suitable title and informative caption. An example of appropriate titling and captioning may be found at https://figshare.com/articles/Table_S2_from_Is_there_a_trade-off_between_peak_performance_and_performance_breadth_across_temperatures_for_aerobic_scope_in_teleost_fishes_/3843624.

Author's Response to Decision Letter for (RSOS-201166.R1)

See Appendix B.

Decision letter (RSOS-201166.R2)

Dear Professor Yoshimura,

I am pleased to inform you that your manuscript entitled "Improving environment drives dynamical change in social game structure" is now accepted for publication in Royal Society Open Science.

If you have not already done so, please remember to make any data sets or code libraries 'live' prior to publication, and update any links as needed when you receive a proof to check - for

instance, from a private 'for review' URL to a publicly accessible 'for publication' URL. It is good practice to also add data sets, code and other digital materials to your reference list.

You can expect to receive a proof of your article in the near future. Please contact the editorial office (openscience@royalsociety.org) and the production office (openscience_proofs@royalsociety.org) to let us know if you are likely to be away from e-mail contact – if you are going to be away, please nominate a co-author (if available) to manage the proofing process, and ensure they are copied into your email to the journal. Due to rapid publication and an extremely tight schedule, if comments are not received, your paper may experience a delay in publication.

on behalf of Prof Miles Padgett (Subject Editor)
openscience@royalsociety.org

Appendix A

Point-by-point response to the Reviewers' comments

We wish to express our appreciation to the Reviewers for insight full comments, which have helped us significantly improve the manuscript. In accordance with the comments, we carefully revised our initial manuscript. Here we present a point-by-point response.

Reviewer comments to Author:

Reviewer: 1

Comments to the Author(s)

*This work establishes a new 2-payer & 2-strategy game. The authors' main claim is that the proposed game has 5 classes, which is more than the conventional 2 by 2 game having 4 game-classes; PD, Chicken, SH and Trivial (non-dilemma game). Their game has PD, Chicken, SH, D-dominant Trivial (that is called Trivial Defection by the authors; which is not appropriate naming: which should be call, as I said, D-dominant Trivial) and C-dominant Trivial (that is called Trivial Cooperation). Although the authors may think it is a significant finding, it seems, at least to me, quite trivial, say, 'literally' trivial honestly. The reason is as below. Referring to the game payoff structure; Fig. 1, depending on the presumed probabilistic parameter; p_0 and p_2 , the order of the diagonal elements; $R (=b*p_2 - c)$ and $P (=b*p_0)$ may change, i.e., both cases; $R > P$ and $R < P$ might be possible. That is why, instead of the conventional 2 by 2 game having only (C-dominant) Trivial, their game has two Trivial sub-classes; splitting it into twofold; C-dominant Trivial (if meeting with $R > P$) and D-dominant Trivial (if meeting with $R < P$). It seems extremely natural, and no surprising at all.*

Theoretically speaking, with the premise of $R > P$, there must be only 4 game classes in symmetric 2 by 2 game, which is clearly proved from mathematical viewpoint. It is because the degree of freedom is 4 in the game. Yet, if one relaxes the constraint of ' $R > P$ ' by allowing the inverse order; ' $R < P$ ', this relaxation unequivocally adds another degree of freedom. Thus, there are five game classes as the authors insisting. From the principle of physics as well, there is nothing to further explain for what they reported here.

REPLY: First of all, it is not the main claim of the present paper that “the proposed game has 5 classes, which is more than the conventional 2 by 2 game having 4 game-classes; PD, CH, SH and Trivial (non-dilemma game)”. In this paper, we divided one of the conventional 4 game-classes (Trivial) into TC and TD, because these two should be necessarily distinguished from our standpoint of discussing cooperative versus defective behavior. Thus, the game has 5 classes. It is still true that the game has 4 classes, as remarked above, if one’s interest lies in the dilemma structure. However, let us repeat that the distinction of TC and TD itself is neither the main claim nor we claim as a significant finding of the present study. To make this point clearer, we revised the manuscript as shown below:

(line 38 in Abstract)

Recently, the weight-lifting game has been shown to cover all five games depending on the success probability of weight lifting, which include the above three games and two trivial cases (all cooperation and all defection; conventionally not distinguished as separate classes).

(line 68-80 in Main Text)

It should be remarked that the two-by-two game is conventionally considered as consisting of four classes because the last two trivial cases (also called C-dominant and D-dominant Trivials [24]) are not distinguished for their common characteristic of no dilemma [24-28]. Here we follow the latter terminology, namely C-dominant Trivial (CT) and D-dominant Trivial (DT) in place of TC and TD, respectively [24,28].

In locally summing so far, I am giving, in a sense, a harsh word to the present contribution. But I wouldn't go so far as saying that I totally deny this work.

Although there is none of substantially big novelty, I can evaluate their work as practically informative. It is because their game extends the conventional 2 by 2 game sphere to broader worlds by introducing another set of probabilistic parameters into the payoff matrix; that are p_0 , p_1 and p_3 . This concept struck me as interesting. What they call ‘weight-lifting game’ sound interesting and catchy to the audience who is interested in the evolutionary game theory. Also their mathematical handling to define those parameters by $p(E, n_c) = E^{\Delta(n_c - 1)}$ looks nice, and quite intrigued, which ensures monotonically increasing function of n_c of which slope increases with increase of E except for

both ends; $E = 0$ and $E = 1$ that are non-sensitive to n_c . The interpretation of E they posed can be ‘environmental’ effect, which is also conceivable.

REPLY: We are very grateful for your appreciation of our study. In fact, introduction of probability p in terms of environmental value E and investigation of the implications thereof are the main contributions of the present paper, while some of the comments are to be directed to the previous work. We revised the manuscript to emphasize the distinctive contribution to the past literature, as indicated below.

(line 65)

While games have been considered separately so far (but see [21]), the weight-lifting game was proposed recently [22,23].

(lines 72-79)

In the weight-lifting game, the probability of success or each player’s payoff depends on the number of cooperators in the game. The uniqueness of this game is that we can cover all the pairwise games in a single game framework by changing the success probabilities, which are considered as input parameters. In the present study, we apply the weight-lifting game model to discuss the evolution of a cooperative society by representing the success probability as a function of a state variable, called environmental value. With this variable, we intend to express the environmental conditions experienced by the society.

To this end, I see this MS can be welcomed to publication only when the revised MS would respond the following suggestions.

#1.

The authors should further deepen their discussion by referring to what I suggested above; 1st and 2nd paragraphs. On the way, they should cite following literature that relating to the universal concept of dilemma strength, why Trivial being called D-dominant Trivial and other theoretically fundamentals;

Evolutionary Games with Sociophysics: Analysis of Traffic Flow and Epidemics, Springer, 2019.1; ISBN: 978-981-13-2769-8

Fundamentals of Evolutionary Game Theory and its Applications, Springer, 2015.10; ISBN: 978-4-431-54961-1

Tanimoto & Sagara; Relationship between dilemma occurrence and the existence of a weakly dominant strategy in a two-player symmetric game, BioSystems 90(1), 105–114, 2007.

Wang et al.; Universal scaling for the dilemma strength in evolutionary games, Physics of Life Reviews 14, 1–30, 2015.

Ito et al.; Scaling the phase-planes of social dilemma strengths shows game-class changes in the five rules governing the evolution of cooperation, Royal Society Open Science, 181085, 2018.

REPLY: We cited the above references in appropriate places.

(lines 68-73)

It should be remarked that the two-by-two game is conventionally considered as consisting of four classes because the last two trivial cases (also called C-dominant and D-dominant Trivials [24]) are not distinguished for their common characteristic of no dilemma [24-28]. Here we follow the latter terminology, namely C-dominant Trivial (CT) and D-dominant Trivial (DT) in place of TC and TD, respectively [24,28].

#2.

As I suggested, their terminology of TD and TC should be replaced with D-dominant Trivial and C-dominant Trivial, of which underpinning was given the two book above from theoretical standpoint.

More precisely, a question of whether it coming to ‘Trivial’ or non ‘Trivial’, let’s say, it coming to ‘Dilemma’ situation is whether or not “Nash Equilibrium is consistent with equal (meaning ‘egalitarian’) social optimal situation”. I suggest the authors should mentioned this point somewhere in Discussion part as well.

REPLY: Following your suggestion, we revised the entire manuscript, i.e., D-dominant Trivial (DT) and C-dominant Trivial (CT) instead of TD and TC of Yamamoto et al. 2019, respectively.

(Line 71)

Here we follow the latter terminology, namely C-dominant Trivial (CT) and D-dominant Trivial (DT) in place of TC and TD, respectively [24,28].

(Lines 361-363)

The colored areas represent all kinds of pairwise games, i.e., the prisoner's dilemma (PD: blue), the chicken game (CH: green), the stag hunt game (SH: red), D-dominant trivial (DT: purple) and C-dominant trivial (CT: yellow).

(Line 367-369)

The colored areas represent all kinds of pairwise games, i.e., the prisoner's dilemma (PD: blue), the chicken game (CH: green), the stag hunt game (SH: red), D-dominant trivial (DT: purple) and C-dominant trivial (CT: yellow).

(Figure 3, Figure 4 and all the expressions of TD and TC in the previous manuscript.)

For the latter point (dilemma situation), we are not sure what is the right solution for now. So we comment this as future research direction, suggesting the current study may be also translated into the dilemma phase plane of GID and RAD of Ito and Tanimoto (2018).

(Line 183)

In future, this dynamical shift in game classes may be translated into the dynamical change in a social dilemma situation (phase plane of GID and RAD [27]) [24]. Then the stability of behavioral phenotypes in humans may be interpreted as the optimal solution under dynamical societal changes.

Reviewer: 2

Comments to the Author(s)

The authors introduce the concept of environmental value which is potentially interesting. However, the authors should be making a stronger effort to make the approach valuable enough. I list some of the reasons below.

The function to capture the related environmental phenomena is not justified enough. The functional is quite specific in its form, taking a similar aspect to those taken by utility functions in economics. The form is not well justified and the authors relies its choice for the sake of simplicity. The authors does not provide a deep exploration about alternative forms and how other forms may affect the current results. The exploration may offer a stronger sense of generality in the discussion section. The lack of a more careful exploration lowers the

impact of the contribution and limits the soundness of the formalism provided.

REPLY: Firstly, the present paper is founded on the conventional framework of game theory, “taking a similar aspect to those taken by utility functions in economics” exactly as pointed out above. In this revision, we made this point clear. (line 88) Indeed, the form we adopted is chosen for the sake of simplicity. However, it is equally possible to adopt another arbitrary form to obtain essentially the same results in so far as it satisfies the conditions stated in the main text. In short, the results are robust and general. To show this explicitly, we supplemented a new Supplementary Material in which we provided corresponding figures for another function.

(line 88)

The present paper is founded on the conventional framework of game theory, taking a similar aspect to those taken by utility functions in economics [35]

(line 139)

It should be remarked that the qualitative behavior of $p(E, n_c)$, but not its particular form ($E^{\delta n_c^{-1}}$), is important for the following results (Supplementary Materials).

The authors introduce the idea of the importance of the context and as I said it is a very interesting perspective but it is just qualitatively addressed in the paper. I wonder if the authors can better (and quantitatively, eventually) connect this with current state of the literature in game theory (some are considering specific groups of individuals while others are considering mobility or space as an aspect). This may indeed mean to better connect the results with empirical contexts and this may allow to offer realistic parameters choice. A deeper literature research for this purpose is encouraged.

REPLY: We made every effort to connect between of a current model and empirical studies of game theory and supplemented the manuscript with literature information. We added eight new references.

(lines 80-97)

Many empirical studies have been carried out in the framework of conventional game theory [29,30]. To account for empirical results from controlled studies, various modifications and generalizations of game theory and expected utility theory have been put forward [29-36]. For example, behavioral game theory takes account for emotions

and limited foresight of average people [33,34]. Recently, a lab-in-the-field experiment was made to study individuals' behavior when facing different situations corresponding to various dyadic games [21]. This study supplied interesting empirical data for simulating societies in the context of game theory. They found that the subjects conform to a limited number of behavioral phenotypes (envious, optimist, pessimist, and trustful) [21]. The present paper is founded on the conventional framework of game theory, taking a similar aspect to those taken by utility functions in economics [35]. However, unlike the traditional studies that focus on the results of an individual game, we study what kind of game is more likely to be provided depending on the environmental state of a dynamically changing society. This type of social investigation has not been made previously. In relation to the current state of the literature in game theory, the success probability of the weight lifting game may be compared with context-related parameters in various specific situations, such as relatedness r in kin selection, the probability of mutual encounter w in direct reciprocity, the probability q of knowing someone's reputation in indirect reciprocity, the average number of neighbors k per individual in network reciprocity, and so on [19,27].

At some point the authors mentions that "the weight-lifting game enables us to investigate them in an unified manner" and that mentioned "games have been considered separately so far". I agree with the first point and concerning the second point I would encourage the authors to look at and related references: Poncela-Casasnovas, J., Gutiérrez-Roig, M., Gracia-Lázaro, C., Vicens, J., Gómez-Gardeñes, J., Perelló, J., ... & Sánchez, A. (2016). Humans display a reduced set of consistent behavioral phenotypes in dyadic games. Science advances, 2(8), e1600451. If I am not mistaken, they directly treat the same games' space and there is a description of existing strategies and there is a data set that can be of interest to the authors.

Thanks for the invaluable information. We corrected the wrong statement by citing the above reference and supplied explanatory sentences (line65, 81)

The abstract ends up with a quite strong sentence: "The current results may offer a solution to unanswered question about the origin of cooperation". The feeling after reading the paper is that the authors are solely able to build a function with two parameters to continuously

travel through the parameters phase diagram where games are all connected. The authors should make stronger efforts in the paper to make the whole results convincing.

REPLY: We removed this line (line 43). Now we understand it is too strong as you pointed out.

Appendix B

Point-by-point response to the Reviewers' comments

We wish to express our appreciation to the Reviewers for insight full comments, which have helped us significantly improve the manuscript. In accordance with the comments, we carefully revised our initial manuscript. Here we present a point-by-point response.

Associate Editor Comments to Author:

The Editors would like you to address the concerns raised regarding both the robustness of the model and also its real-world applications – these are reasonable concerns from the reviewer, and we would like you to include a paragraph or two to provide more exposition here. If you provide this in a revised paper, as well as your response to referees document, the paper may be accepted – but this will be subject to you satisfactorily engaging with these concerns.

REPLY: Thank you for your accept decision under minor revision. We did the best to explain the robustness of our theory and the future perspectives on its real-world applications. We present a point-by-point response.

Reviewer comments to Author:

Reviewer: 1

Comments to the Author(s)

The revised MS seems to be acceptable for publication.

REPLY: Thank you for your great decision.

Reviewer: 2

Comments to the Author(s)

The authors have answered to most of my concerns and I thank them for this effort.

1) I however still find the paper very disconnected to empirical features or specific situations. The authors have not shown any means to anticipate how the model could become valuable to better interpret real-world situations. There is still no clue on how the model can take real world observation to calibrate somehow the model.

REPLY: Thank you for your fundamentally important comments. It is the most important in science, since no theory is valid unless empirically tested. In this report, as a first trial of the theory, we illustrate the model necessity with numerous historical facts of rise-and-fall in human culture, as a conceptual way. In this current revision, we provide some future aspects of the model verification along with the future.

Page 15-16 lines 256-265 (a paragraph)

Inserted Text:

It should be very important to verify the model in science, since no theory is valid unless empirically tested. In this report, as a first attempt of the theory, we illustrated the model necessity (assumption), that is, the probability of success is an increasing function of historical changes over time. This corresponds to the numerous historical facts of rise-and-fall in human culture, as a conceptual way. The direct verification of the model is, however, very difficult because such historical rise-and-fall is difficult to quantify. The verification of the current model is a future step. One possible method is to compare the human responses against the weight-lifting game before and after a radical societal change over a short time period, e.g., before and after the COVID-19 pandemic. Another similar approach is to compare various societies under different conditions, e.g., the comparison of the responses of people among various countries.

2) Concerning the robustness of the model, the authors should be better explaining why the model is robust enough and why the alternative model supports this statement. The authors have included an obscure sentence in the main paper in relation to these issues and this fact needs further clarification.

REPLY: Thank you for your valuable comments about the robustness. We add a few lines to explain why we think our results is robust and the current results are general in this sense. Main point is that the robustness of the results is originated from the assumption of the probability increasing with time from near 0 to 1 (unity). The interpretation of the robustness was not clear in the previous version, simply because we do not know why this necessity should be interpreted as a social change over time. The assumption of increasing probability (0 to 1) can be relaxed in a certain cultural history and we then get a totally different result. We included these ideas in the revision below.

Page 10 lines 159-166 (a paragraph)

Inserted Text:

The results of the proposed model are robust because these results originate from the general assumption that the probability of success increases from near 0 to 1 (unity) (Figs. 2, 3, Supplementary figs. S1, S3, S5). Note that the increasing probability is expressed as a left-to-right trajectory in Figure 3(c), 3(f) and Supplementary figs S2(c), S4(c), S6(c). This probability assumption simply expresses that a wealthy society has more success than a poor society. The assumption of increasing probability (0 to 1) can be violated in some irregular societal conditions. We should also relax this probability condition when we simply compare various present societies simultaneously.